# *CiXTH29* and *CiLEA4* Role in Water Stress Tolerance in *Cichorium intybus* Varieties

**DOI:** 10.3390/biology12030444

**Published:** 2023-03-13

**Authors:** Monica De Caroli, Patrizia Rampino, Lorenzo M. Curci, Gabriele Pecatelli, Sara Carrozzo, Gabriella Piro

**Affiliations:** 1Department of Biological and Environmental Sciences and Technologies, University of Salento, 73100 Lecce, Italy; 2NBCF National Biodiversity Future Center, 90133 Palermo, Italy

**Keywords:** *CiXTH29*, *CiLEA4*, drought stress, drought markers, xyloglucans

## Abstract

**Simple Summary:**

Climatic factors, such as drought and high temperatures, drastically affect the productivity of cultivated plants. The identification of plants showing stress resistance characteristics, as well as the identification of morphological or molecular traits associated with drought tolerance, is extremely important, because progressive climate change is expected to further reduce water availability, with dramatic impacts on crop productivity. We investigated the drought stress responses of six varieties of *Cichorium intybus* L., i.e., Selvatica, Zuccherina di Trieste, Brindisina, Esportazione, Rossa Italiana and Galatina. Our results highlight that Selvatica and Zuccherina di Trieste varieties better tolerate a drought stress condition than other varieties, and this was used in the identification of potential markers for drought stress tolerance.

**Abstract:**

Drought causes massive crop quality and yield losses. Limiting the adverse effects of water deficits on crop yield is an urgent goal for a more sustainable agriculture. With this aim, six chicory varieties were subjected to drought conditions during seed germination and at the six week-old plant growth stage, in order to identify some morphological and/or molecular markers of drought resistance. Selvatica, Zuccherina di Trieste and Galatina varieties, with a high vegetative development, showed a major germination index, greater seedling development (6 days of growth) and a greater dehydration resistance (6 weeks of growth plus 10 days without water) than the other ones (Brindisina, Esportazione and Rossa Italiana). Due to the reported involvement, in the abiotic stress response, of xyloglucan endotransglucosylase/hydrolases (XTHs) and late embryogenesis abundant (LEA) multigene families, *XTH29* and *LEA4* expression profiles were investigated under stress conditions for all analyzed chicory varieties. We showed evidence that chicory varieties with high *CiXTH29* and *CiLEA4* basal expression and vegetative development levels better tolerate drought stress conditions than varieties that show overexpression of the two genes only in response to drought. Other specific morphological traits characterized almost all chicory varieties during dehydration, i.e., the appearance of lysigen cavities and a general increase of the amount of xyloglucans in the cell walls of bundle xylem vessels. Our results highlighted that high *CiXTH29* and *CiLEA4* basal expression, associated with a high level of vegetative growth, is a potential marker for drought stress tolerance.

## 1. Introduction

Global warming and reduced rainfall due to global climate change cause extended drought periods throughout the globe. Every year, rainfed agriculture suffers output losses due to severe water deficit adverse effects [1]. Drought is one of the major stressors strongly affecting plant growth, development and crop productivity [2,3]. To adapt to environmental stresses, plants have evolved a range of morphological, physiological, biochemical and molecular cooperative mechanisms modulated by complex signaling networks [4]. Differential expressed genes have been identified in tolerant and susceptible maize seedlings, mainly associated with the cytoskeleton, glycolysis/gluconeogenesis, carbon fixation pathways, transport, osmotic regulation, drought avoidance, ROS scavengers, transcriptional factors, photosynthesis, histone acetylation and cell wall modification [5,6]. Plant cell wall integrity maintenance plays a key role in stress perception and stress responses in plants [7,8,9]. Although many questions remain to be addressed in this process, several cell wall sensors involved in cell wall integrity monitoring have been identified [10,11]. Drought stress affects the plant cell water potential and turgor pressure, causing cell wall modifications that involve the molecular organization and interaction of the wall polymers. The increase in cell wall elasticity (CWE) is one of the first pieces of evidence regarding the physiological mechanisms of adaptation to water stress, and CWE seems to be important for maintaining cell turgor and symplast volume [12,13,14,15]. Furthermore, as a drought stress response, different plant species show an altered cell wall biosynthesis, generally consisting of a decreased level of cellulose content, as reported in grape [16], chickpea epicotyls [17] and wheat [18,19]. In poplar, chemical analyses revealed that the total production of cellulose and xylans was significantly reduced under drought conditions; transcriptomic analysis in young shoots supported these results, showing a drastic change in the expression of cell-wall-related genes, with a significant decrease in the expression those involved in cellulose and hemicellulose biosynthesis [20]. A detailed analysis of genes involved in cell wall metabolism has been reported in *Arabidopsis* seedlings, showing a generally reduced gene expression during soil water deficits, in particular, of genes promoting cell wall modification, such as expansins, pectin esterases and xyloglucan endotransglucosylase/hydrolases (XTHs) [21,22]. The involvement of expansins and XTHs in the remodeling of the wall has also been confirmed in the perspective of the new cell wall models that emphasize the binding between xyloglucans, the major hemicellulosic polysaccharides of dicot plants, and cellulose microfibrils at limited sites named ‘hot spots’ that can be modulated to control wall expansion [11,23,24]. The involvement of XTHs in the drought stress adaptation has been evidenced in durum wheat cultivars differing in their stress tolerances [25], in *Arabidopsis* and tomato seedlings overexpressing the *Capsicum XTH3* gene [26,27] and in soybeans overexpressing *GmXTH1* [28]. In addition, in *Arabidopsis*, three members of the XTH family, XTH11, XTH33 and XTH29, have been reported to be differently involved in the drought stress response in the roots and aerial parts of seedlings. Significantly, XTH29, the only isoform among the three that follows an unconventional protein secretion, was highly up-regulated with respect to the others, undergoing a conventional secretion [29].

The cell wall is certainly the first compartment involved in the perception and response to stress, but several metabolic rearrangements occur during drought plant adaptation, and different metabolites have been identified as protective molecules involved in the short- or long-term drought responses. Several secondary metabolites belong to the class of short-term stress adaptations, such as glycine betaine, species-specific for sugar beet, spinach and barley, or proline; the increases in proline seem to be the most conserved drought stress response for many plant species [30]. The increased expression of drought-specific genes and the biosynthesis of drought-protective proteins represent, instead, the most significative responses in the drought long-term adaptation class; chaperonins and late embryogenesis abundant (LEA) proteins are the classes of proteins mainly involved in these long-term responses. LEA proteins are particularly associated with stressors that include dehydration, or those that occur during some plant development stages involving water limitation, such as during seed, pollen grain, shoot and root development [31]. LEAs are widely distributed proteins in the plant kingdom, from algae to angiosperms [31], and, inside the cell, they are localized predominantly in the cytosol, but their presence has been highlighted also in the chloroplast, mitochondria, protein and lipid bodies, plasmodesmata and nucleus [32]. The in vivo activity for most of LEA proteins remains unknown, but their positive correlation with abiotic stress tolerance has been inferred from their increased expression in response to abiotic stress, as well as from the increased stress tolerance in *LEA*-overexpressed transgenic plants [33]. LEA4 is the most abundant molecule in the LEA family of *Arabidopsis* [34], and the silencing of *AtLEA4* has been proven to be sufficient to cause water deficit susceptibility [35]. LEA4 is characterized by a large number of α-helices, with a strong hydrophilicity, which favors cells to enhance the water absorption under drought stress [34]. As reported for other *LEA* family members, transgenic plants overexpressing *LEA4* show a high resistance to salt and drought stress [36].

Chicory (*Cichorium intybus* L.) is an Asteraceae herbaceous plant that is native to Mediterranean and middle Asian regions [37], and is now widely cultivated in different temperate areas of Europe and America, due to its ability to adapt to a range of different climate and soil conditions. Chicory has been used since ancient times as vegetable crops and medicinal plants and, over time, it has been used for many commercial uses, including as a source of inulin, a fodder plant, a coffee substitute/additive and for ethanol production [38]. Its biochemical composition has been characterized by several studies supporting its health benefits and use as source of phytochemicals [39,40,41]. Chicory is a plant particularly appreciated not only as a traditional Italian food product, but also for its antioxidant properties, characteristics that have increased demand for it in terms of quantity and quality. Several studies carried out on chicory have led to the identification and isolation of a wide variety of phytochemicals, including anthocyanins, coumarins, flavonoids, fructans and sesquiterpene lactones [41]. All the organs of the chicory plant are used as food, but the most frequently consumed organs are the rosette basal leaves, eaten raw or cooked. Several pieces of evidence support the positive effect of water availability on chicory yield [42,43]; furthermore, the reduced water availability imposes morphological changes on chicory plants, such as in the canopy structure, and this negatively affects biomass accumulation [15,16]. In a recent study, chicory plants submitted to the stressor of rainfed conditions showed an increase in carotenoid and tocopherol contents, suggesting thee positive effect of moderate rainfed drought stress on the plants [44].

The main objective of this study was to explore the possibility that plant morphology and stress gene expression within chicory varieties might help to understand some new aspects of drought stress tolerance in plants. For this reason, we studied the response to drought stress in wild (Selvatica) and cultivated (Zuccherina di Trieste, Brindisina, Esportazione, Rossa Italiana and Galatina) chicory varieties. Moderate and severe drought stress was administered during germination, while the supply of water was suspended in six week-old plants for 10 days. After morphological observation of drought stress impacts on six week-old chicory plants, *CiXTH29* and *CiLEA4* gene expression was performed in control and water deficit conditions. Our results could lead to the identification of some morphological and molecular markers of drought resistance that could be beneficial for future improvements in variety preservation strategies or chicory genetic breeding, in order to promote better drought stress tolerance.

## 2. Materials and Methods

### 2.1. Plant Material, Seed Germination and Seedling Drought Treatment

*Cichorium intybus* L. seeds of the Selvatica, Esportazione, Rossa Italiana and Galatina varieties were provided by the Larosa Emanuele Sementi company (Trani, Italy); Zuccherina di Trieste variety was provided by the Ferritalia company (Padova, Italy), and Brindisina variety was provided by the Franchi Sementi company (Grassobio Bergamo, Italy). Chicory seeds were soaked for 2–3 h in running tap water, surface-sterilized with 0.6% (*v/v*) commercial bleach solution for 10 min, then germinated and grown, at 22 °C for 6 days in the dark, on Petri dishes (Ø = 9 cm) containing two layers of Whatman No. 1 filter paper (Whatman Ltd., Maidstone, UK) embedded with either 5 mL of distilled water, or 5% or 15% PEG-6000 solution [28]. High-molecular-weight PEG reduces water potential primarily by matric forces, rather than osmotic forces, and it is not able to penetrate the pores of plant cell walls, thus causing cytorrhysis (withdrawal of water from and shrinkage of both the cell wall and protoplast), rather than plasmolysis, where only the protoplast loses water and may separate from the cell wall. For these reasons, treatment with high-molecular-weight PEG is more similar to soil-drying than the osmotic stress imposed by low-molecular-weight solutes [45]. Further, PEG-6000, a non-penetrating osmotic agent, lowers water potential, causing a water stress condition corresponding to 0 (mock), −0.1 MPa ψ_W_ (5%) fir moderate drought stress, and −0.4 MPa ψ_W_ (15%) for severe drought stress [46]. Each experiment was repeated 5 times, sowing ten seeds per plate, with a total of 50 seeds for each variety and treatment. After 6 days, the germination index (GI) was calculated as the percentage of germinated, healthy seedling among the total number of tested seeds; the 6 day-old seedling root length and aerial part (epicotyl and first leaflets) height were measured with a ruler under a stereomicroscope (Stemi 508, Zeiss, Oberkochen, Germany).

For RNA extraction, RT-qPCR and immunolocalization analyses, seeds of the different varieties were sown in plastic pots filled with wet soil, with 100 mL of water, and transferred in a growth chamber (22 °C temperature, 60% humidity, 25 μE light intensity, 16/8 photoperiod). Every three days, 100 mL of water was added to the growing seedling. After 6 weeks, the plants were divided into control and stressed groups. Drought stress was induced by withholding the irrigation for a further 10 days of growth, while the control plants were grown in parallel, and were regularly watered. At the end of the stress treatment, basal rosette leaves were used for the subsequent analyses. Furthermore, representative mock and drought stress seedlings (6 days old) and plants (6 weeks old, plus 10 days with or without watering) of each variety were selected and photographed.

### 2.2. RNA Extraction, Cloning and Sequence Analysis

Total RNA isolation and cDNA synthesis were performed as reported by De Paolis et al. [47]. Total RNA was isolated from basal rosette leaves of 6 week-old chicory plants (Selvatica, Zuccherina di Trieste, Brindisina, Esportazione, Rossa Italiana, Galatina) subjected, or not, to drought stress for a further 10 days. To amplify the nucleotide sequence of chicory *XTH29*, 100 ng of cDNA was used as a template in the presence of 10 pmol of *At*XTH29-3′clonefor and *At*XTH29-3′clonerev primers, designed based on the *XTH29* gene of *Arabidopsis* (Appendix A), by using 2.5 mM of each dNTP and 1.5 U of TaKaRa Ex TaqTM (TAKARA BIOTECHNOLOGY, Kusatsu, Japan) in 50 µL of the buffer solution, as indicated by the supplier. After a denaturation step at 95 °C for 2 min, *CiXTH29* amplification reactions were carried out for 35 cycles for 30 s at 95 °C, for 30 s at the annealing temperature of 57 °C and for 1 min at 72 °C. A final elongation step was run at 72 °C for 10 min. The amplification products were cloned into pGEM-T Easy (Promega, Milano, Italy) for subsequent sequencing (Eurofins Genomics, Anzinger, Germany). The sequence analysis and comparison with other sequences were performed using the NCBI BLAST web tool (https://blast.ncbi.nlm.nih.gov, accessed on 15 December 2021) (Appendix A). The deduced amino acid sequence of *Ci*XTH29 was obtained with the EXPASY web tool (https://web.expasy.org/translate, accessed on 15 December 2021) and was analyzed through the BLASTP (https://blast.ncbi.nlm.nih.gov, accessed on 15 December 2021) to define the identity percentage among the other sequences present in BLAST database (Appendix A). Alignment of the *Ci*XTH29 sequence with *At*XTH29 was conducted using the EMBL-EBI web tool (Clustal Omega, https://ebi.ac.uk/Tool/msa/clustalo/, accessed on 15 December 2021).

### 2.3. RT-qPCR Analysis for CiXTH29 and CiLEA4 Gene Expression

RT-qPCR analysis was performed on equal amounts of cDNA from control and stressed samples, using SYBR Green fluorescent detection in a CFX96 Real-Time System Cycler (Bio-Rad, Hercules, CA, USA), with three biological replicates per variety and treatment. The primer sequences used are reported in Appendix A. *β-tubulin* (*CiTUB*, KP752084) (PCR program: 10 min at 95 °C; 40 cycles of 15 s at 95 °C; 20 s at 57 °C) and *actin* (*CiACT*, KP752080) (PCR program: 10 min at 95 °C; 40 cycles of 15 s at 95 °C; 20 s at 59 °C) were tested as reference genes [48]. For *CiXTH29*, the PCR program was as follows: 10 min at 95 °C; 40 cycles of 15 s at 95 °C; 20 s at 57 °C. For *CiLEA4*, primer pair (Appendix A) previously reported for *CsLEA4* [49] was used for RT-qPCR analysis, with the following program: 10 min at 95 °C; 40 cycles of 15 s at 95 °C; 20 s at 57 °C. The specificity of PCR products was checked in a melting-curve test. The PCR product was sequenced and compared with other sequences using the NCBI BLAST web tool (accessed on 15 December 2021) (Appendix A) confirming that *CiLEA4’*s identity with *AtLEA4* was 98%. Gene expression differences between mock and stressed samples were considered significant when gene expression was at least doubled (greater than or equal to two-fold up-regulation) or halved (less than or equal to two-fold downregulation), according to [50].

### 2.4. Leaf Sections and Xyloglucan Immunolocalization

From 6 week-old chicory plants subjected, or not, to drought stress for a further 10 days, basal leaf segments of 5 mm width and 5 mm height, measured using a caliber, were accurately cut with a razor blade and immediately fixed in cold (4 °C) FAA solution (50% ethanol, 0.5% glacial acetic acid, 10% formaldehyde (37–40%), water up to the final volume) overnight. Fixed samples were dehydrated and then included in paraffin as described in De Caroli et al. [51]. Paraffin blocks were sectioned to a thickness of 15 μm with a LEICA RM2155 microtome. For bright field images, the sections were mounted on slides, and images were acquired with 10X objective with a Zeiss LSM710 microscope. The images were then reconstructed using Adobe Photoshop CC (64 Bit). For immunolocalization analyses, the sections were blocked with 5% (*w/v*) skimmed milk in PBS for 30 min and incubated with primary antibodies (LM24, Kerafast, Boston, MA, USA) at 1:10 dilution for 1 h. The slides were washed three times with PBS for 10 min, probed with secondary antibodies (anti-rat antibodies conjugated to AlexaFluor488; Life technologies, Carlsbad, CA, USA) at a 1:250 dilution for 1 h, and were then washed three times with PBS for 10 min. The observations were performed with a Zeiss LSM710 microscope. To detect AlexaFluor488 fluorescence, a 488 nm argon ion laser line was used, and the emission was recorded with the 505–530 nm filter set; after He–Ne laser excitation at 543 nm, the tissue autofluorescence was detected with the filter set for tetramethylrhodamine isothiocyanate (TRITC; >650 nm). The power of each laser line, the gain and the offset were identical for each experiment so that the images were comparable. For fluorescence quantification, the Profile tool in ZEN2012 program was used. Using standardized settings, the ratio of green to blue fluorescence pixels was used as an index of fluorescence.

### 2.5. Statistical Analysis

Statistical analyses were based on the Student’s *t*-test. Statistical comparisons were performed using SigmaStat software, version 11.0 (Systat Software Inc., Chicago, IL, USA)

## 3. Results

### 3.1. Seed Germination and Chicory Seedling Growth under Drought Stress

Six days after sowing, the seeds of the chicory varieties showed a different germination index (GI), as well as a different ability to germinate under water stress conditions, determined by 5% and 15% PEG-6000, respectively, miming the moderate and severe drought conditions. In the absence of PEG-6000 (mock), the average GI for all varieties was 80.6% (ranging from 97% to 60%), which dropped to 73.1% with 5% PEG-6000 solution, and to 58.8% with 15% PEG-6000 solution (Table 1). Selvatica and Zuccherina varieties showed a significant decrease of GI only with the 15% PEG-6000 solution, while all other varieties had significantly decreased GIs with both the 5% and 15% PEG-6000 solutions. The drought stress imposed by the concentration of 5% PEG-6000 determined a slight decrease in the GI value of the Brindisina (26.3%), Esportazione (15.5%), Rossa Italiana (7.13%) and Galatina (6.5%) chicories, with this concentration representing moderate drought conditions. For the 15% PEG-6000 treatment, the GI decreased slightly in Selvatica (12.9%), Zuccherina (14.8%), Esportazione (25.25%) and Galatina (17.3%), while it decreased markedly in Brindisina (45.5%) and Rossa Italiana (49.9%) varieties.

After 6 days of seed germination, the Selvatica variety showed higher average growth values than the other varieties for both the root and aerial parts (epicotyl and first leaflets), while the Galatina variety appeared to be the less developed one (Figure 1). In the aerial portion, an imposed condition of moderate drought stress (5% PEG-6000) caused a growth inhibition between 46% and 58% for the Selvatica, Zuccherina di Trieste and Galatina varieties; these values reached between 80% and 82% under more severe stress conditions (15% PEG-6000). For the Brindisina, Esportazione and Rossa Italiana seedlings, the growth inhibition showed higher values than the varieties listed above, being between 62 and 65% and 86 and 88% under moderate and severe drought stress, respectively. In the roots, which grew less than the aerial part in all varieties, the percentages of inhibition reflected the trend already evidenced in the aerial part, with values between 35 and 48% and 75 and 80% for Selvatica, Zuccherina di Trieste and Galatina varieties, and values of 51 and 61% and 86 and 88% for Brindisina, Esportazione and Rossa Italiana varieties, in drought stress conditions imposed by 5% and 15% PEG-6000, respectively (Table 2).

### 3.2. Phenotypic Analysis of Chicory Plants under 10 Days of Dehydration

We compared the capability of the analyzed chicory varieties to respond to dehydration; for this purpose, 6 week-old chicory plants, grown in watered soil, were left for a further 10 days with or without watering. As evidenced in Figure 2, the chicory varieties had a different phenotype, mainly related to the different development rates of the basal rosette leaves, and, after 10 days of water deficit, they showed generally different degrees of wilting, phenotypically demonstrating the actual water deficit stress. Under control conditions, the Selvatica, Zuccherina di Trieste, Brindisina and Esportazione varieties showed a greater vegetative growth than the Galatina and Rossa Italiana plants. After the dehydration period, all the varieties showed a less developed phenotype than their respective controls; the leaves were, in general, moderately drooping, wilting, curled and slightly yellowing. In addition, in the Galatina and Rossa Italiana varieties, all the basal leaves appeared to be significantly shriveled, while in the more vegetatively developed varieties (Selvatica, Zuccherina di Trieste, Brindisina and Esportazione), some leaves still appeared turgid (Figure 2).

The detailed cellular phenotypes of the different plants were analyzed by comparing thin transverse sections carried out at the central vein level of each chicory leaf, all showing a typical dorsoventral organization (Figure 3). Almost all varieties showed more expanded parenchyma cells in the median position of the single central vein (midvein); only the Brindisina and Galatina varieties showed lysigenous cavities in the corresponding area. Beyond this specific characteristic, no other significant morphological variations were observed in the parenchyma tissue, in which the bundles of different size were immersed. The bundles at the level of the central vein were mostly well differentiated in all varieties, and have a developed cap of sclerenchyma fibers at both the phloem and xylem levels. The number of the vascular bundles in the midvein was different in the different varieties, and was higher in the more developed ones. The leaves of Selvatica, Brindisina and Esportazione varieties had five bundles, with dimensions that progressively decreased from the center to the periphery of the midvein. The Zuccherina di Trieste variety showed three larger bundles and, laterally, two small very reduced bundles; the central rib of the Rossa Italiana and Galatina plants was characterized by only three bundles. The leaf lamina of all the varieties showed bundles of reduced dimensions, and was made up of few elements that were not fully differentiated (Appendix A). After a period of water stress, no variations in the number of bundles were evident. Furthermore, the expanded parenchyma cells gave rise to large lysigenous cavities in all the stressed varieties, with the exception of the Rossa Italiana, which still showed highly expanded cells. In the Brindisina variety, the lysigenous lacunae, well visible in the controls, become a single large cavity (Figure 3).

### 3.3. CiXTH29 and CiLEA4 Gene Expression in Chicory Varieties

To analyze the *XTH29* expression profile in chicory varieties, a nucleotide sequence of about 700 bp was amplified from basal rosette leaf cDNA of 6 week-old chicory plants, grown for a further 10 days with (mock) or without watering (drought stress condition), using a primer couple based on the *XTH29* nucleotide sequence of *Arabidopsis* (Appendix A). The 700 bp amplimer was checked by sequencing (Eurofins Genomics, Anzinger, Germany). The deduced amino acid sequence (Expasy Translate Tool) was analyzed through the Basic Local Alignment Tool (BLASTP) with all the other sequences present in the BLAST database. A high (99.13%) identity with an E value of 3 × 10^−169^ was observed between *Ci*XTH29 and *At*XTH29, sequence ID: NP 193634.1 (Figure 4 and Appendix A). XTH family genes contain a typical catalytic enzymatic reaction motif (HDEIDFEFLG), in which the first glutamic acid residue (E) is an affinity site and the second is a proton donor [52,53], as evidenced in Figure 4. The partial sequence of *Ci*XTH29, reported for the first time, was submitted to the NCBI database (submission number: OQ572736).

To identify stable reference genes in chicory varieties, we tested the possibility of using chicory *β*-*tubulin2* (*CiTub2*) and *actin2* (*CiAct2*) as reference genes in real time- quantitative PCR (RT-qPCR) analyses. The analyses were performed with primers previously [48] used in *C. intybus* for the amplification of *actin2* and *β-tubulin2* (Appendix A). All primer couples were tested on mRNA samples isolated from the basal rosette leaves of 6 week-old chicory varieties, grown for a further 10 days with (mock) or without watering (drought stress condition); the mean and standard deviation of the Cycle threshold (Ct) values for each putative reference gene were analyzed and compared (Figure 5, Appendix A). The results of the two reference genes’ amplification displayed a range of transcription levels across all analyzed samples, with average Ct values ranging from 23 to 28 (Figure 5, Appendix A). Since gene expression levels are negatively correlated with Ct values, the amplification product obtained with *CiAct2* primer pairs exhibited the highest abundance, with the lowest mean Ct value of 23.51 ± 0.47 observed in Rossa Italiana plants under the mock condition, while the highest mean Ct value was 25.36 ± 0.15, observed in the drought-stressed Esportazione variety.

The lowest Ct value for *CiTub2* amplimers was 23.99 *±* 0.19 in the basal rosette of Rossa Italiana mock plants, and the highest Ct value was 28.18 *±* 0.50 in the stressed Zuccherina di Trieste variety, showing a wider range, among the Ct values, of *CiTub2* amplimers than that obtained with the *CiAct2* primer pair. In fact, in addition to the abundance of product amplified by the *CiAct2* primer pair, *CiAct2* showed the lowest coefficient of variance (CV) with respect to that obtained with the *CiTub2* primer pair, although the two genes, under both mock and stressed conditions, had a CV below 0.1, which is a negligible variation under stress conditions [54] (Appendix A). All the data showed that the *CiAct2* primers were eligible for the amplification of an appropriate reference gene for chicory in all varieties and experimental conditions of this research.

The chicory *XTH29* primer pair (Appendix A) was used to amplify the *XTH29* gene from chicory cDNA for RT-qPCR analyses. The primers based on *CiXTH29* gave amplification products for all chicory varieties, and the size of the PCR products was visualized by electrophoresis on 2.0% agarose gel, confirming the production of a single amplicon (Appendix A). The specificity of each primer pair was verified by melting curve analysis, which revealed that the gene had a single amplification peak. Furthermore, the cloned PCR products were sequenced (Eurofins Genomics, Anzinger, Germany). The results confirmed the specificity of primers, since 100% identity was found for the nucleotide sequence of *AtXTH29* for all queried varieties (Appendix A). In mock conditions, the RT-qPCR analysis revealed the highest *CiXTH29* expression levels in Selvatica and Zuccherina di Trieste (0.00340 *±* 0.00040 and 0.00270 *±* 0.00040, mRNA values reported as 2^−∆Cq^
*±* SD respectively), intermediate values in Rossa Italiana and Galatina (0.00111 *±* 0.00021 and 0.00147 *±* 0.00038) and the lowest in Brindisina and Esportazione (0.00051 *±* 0.00009 and 0.00098 *±* 0.00060, Appendix A and Figure 6), although the latter had a large variation that also overlapped with the gene expression observed in Rossa and Galatina varieties. The 10 day-long drought treatment showed slight changes in *CiXTH29* expression in Selvatica and Zuccherina di Trieste (log_2_ of Foldchange, log_2_FC, approximately around zero: −0.58 *±* 0.11 and 0.51 *±* 0.16), while statistically significant changes were observed in Galatina (log_2_FC: 2.13 *±* 0.39), Brindisina (log_2_FC: 3.06 *±* 0.23), Esportazione (log_2_FC: 2.59 *±* 0.75) and Rossa Italiana (log_2_FC: 2.26 *±* 0.16). Note that the Selvatica variety was the only variety that showed reduced gene expression after the drought treatment, even if the values are not statistically significant (Figure 6).

Similar to the *CiXTH29* expression pattern, the *CiLEA4* relative basal amount had the highest value in Selvatica plants (0.0040 *±* 0.0010, Appendix A and Figure 7), followed by Zuccherina di Trieste and Galatina varieties (0.0017 *±* 0.0004 and 0.0010 *±* 0.0005, Appendix A and Figure 4). Furthermore, after 10 days of water deficit, *LEA4* expression remained almost constant in Selvatica, Zuccherina di Trieste and Galatina (log_2_FC: 0.32 *±* 0.30; log_2_FC: −0.32 *±* 0.21; log_2_FC: 1.25 *±* 0.13) varieties, while gene overexpression was evident in Brindisina, Esportazione and Rossa Italiana chicories (log_2_FC: 2.65 *±* 0.27; log_2_FC: 2.18 *±* 0.12; log_2_FC: 2.10 *±* 0.05) (Figure 7).

### 3.4. Xyloglucan Quantification through Antibody Assay

Monoclonal antibodies have shown to be highly sensitive molecular tools for the detection of cell wall polysaccharides. We used the LM24 xyloglucan monoclonal antibody [56] to detect the presence of xyloglucans in the cell walls of the histological elements of the largest bundle present in the leaf midvein of chicory varieties grown in water (Mock) and without watering (Drought). Thin sections obtained at the level of the central vein were immunolabeled with LM24, and the fluorescence index was measured as the ratio between the antibody fluorescence and the autofluorescence of the analyzed tissues. In all chicory varieties, the xyloglucans probed by LM24 were detected in the cell wall of xylem vessels (Figure 8). The highest detection of xyloglucans was found in the Brindisina variety grown under control conditions or subjected to ten days of water deficit; under this condition, an increase in xyloglucan in the xylem vessels equal to 170% was observed. The xyloglucan content values in the Selvatica and Zuccherina di Trieste varieties were lower than in the Brindisina one, both in mock and drought conditions, and sharply increased, respectively, to 112% and 106% during water stress. The Esportazione and Rossa Italiana varieties did not show significant changes in xyloglucan content after 10 days of water deficit, while in the Galatina bundle leaves, the detection of xyloglucans increased by approximately 44% after drought stress.

## 4. Discussion

In this study, we focused our attention on the expression of *CiXTH29* and *CiLEA4* and on their role in the drought stress response in six chicory varieties, i.e., Selvatica, Zuccherina di Trieste, Brindisina, Esportazione, Rossa Italiana and Galatina. The analyzed chicory varieties show different phenotypes between the Selvatica, Zuccherina di Trieste, Brindisina and Esportazione chicory plants, characterized by greater vegetative development compared to the Rossa Italiana and Galatina varieties. Within these two groups, with high or low growth, some of these varieties show a better ability to adapt to a period of moderate (5% PEG-6000) or severe (15% PEG-6000) drought stress during seed germination and the initial phase of growth. In the first group, the Selvatica, Zuccherina di Trieste and Brindisina varieties exhibited similar characteristics but, among the three, the Selvatica chicory shows a higher GI, a greater development of the roots and aerial parts during short-term growth (6 days, Figure 1) and a greater development of the basal rosette during long-term growth (6 weeks plus 10 days, Figure 2), compared to the others. In addition, Selvatica is the variety that better adapts to the condition of water stress. On the contrary, the Brindisina chicory is the variety that most suffers from water stress, as highlighted in almost all analyzed parameters. In the second group, the Galatina variety shows generally lower growth parameters than the other varieties, but these values are proportionally less affected under stress conditions.

Germination is drastically affected by soil water content, and the percentage of seed germination represents a significative indication of a plant’s drought tolerance; in fact, drought stress greatly affects seed germination, but the response intensity and the occurrence of the harmful effects of stress depend on the species [57]. The Selvatica and Zuccherina di Trieste varieties show GI values that are still quite high, even under severe drought stress conditions; interestingly, the Galatina variety, even when starting from a low initial GI, shows percentages of germination inhibition comparable to the more resistant varieties of the first group. These results imply that the seeds of the six chicory varieties have a different water demand for germination. Likely, these different PEG-6000 responses of chicory seeds reflect the different ability of the analyzed varieties to tolerate drought stress during the germination stage. In addition to the germination data, the seedling’s short-term growth (6 days) under drought stress conditions differentiates the group containing the Selvatica, Zuccherina di Trieste and Galatina varieties from the group containing Brindisina, Esportazione and Rossa Italiana, with the former showing an inhibition of the growth of roots and the aerial parts that is less affected by a condition of mild and severe stress compared to the latter, which proves to be much more sensitive to drought.

After a dehydration period of 10 days imposed on the 6 week-old chicory plants, significant differences in the overall phenotypes of the analyzed chicory varieties were observed, mainly related to a general reducing, wilting and yellowing of the basal rosette. A greater dehydration of the basal rosette leaves in the Rossa Italiana and Galatina varieties in comparison to the other ones occurred. In soybean plants subjected to water deficits, a general reduction in the leaf lamina has been reported, associated with a greater presence of intercellular spaces hypothesized to make it possible for cells to remain more juxtaposed [58]. For the midvein bundles of the analyzed chicory varieties, we found variations neither in xylem element differentiation, nor in the amount of sclerenchyma fibers, always present close to the phloem and xylem. On the contrary, an increased distribution of sclerenchymatic cells has been reported in the phloem of stems and leaves of soybean plants suffering from drought stress, highlighting that sclerenchyma tissue provides an adaptive advantage against drought stress [58,59]. With the exception of Brindisina and Galatina varieties, showing lysigen cavities in the median position of the midvein, all the other varieties present a compact mesophyll with more expanded parenchyma cells. However, during dehydration, the lysigen cavities increase in size in Brindisina and Galatina varieties and, more significantly, appear to have a large size than all the other varieties. The presence of lysigenous aerenchyma in leaves decreases the symplastic and apoplastic movement of water to the guard cells, the main site of transpiration [60]. This may be one of the numerous strategies used to respond to water stress.

The xyloglucan endotransglucosylase/hydrolases (XTHs) are cell-wall-modifying enzymes, having a pivotal role in maintaining the integrity and strength of the cell wall under normal and stressful environments. Increased XTH transcript levels have been observed in plants exposed to multiple stresses; thus, they represent a cell wall gene family related to abiotic stress response in plant genomes [61,62,63]. We have previously mentioned the involvement of *XTH29* in the rapid response to water and heat stress in *Arabidopsis* [28]. Due to the different adaptive capabilities under conditions of drought stress in the analyzed chicory varieties, we evaluated the expression level of *XTH29* in the six chicory varieties under water stress. At the same time, we also analyzed the expression of *LEA4*, a member of the LEA protein family, a ubiquitous protein functioning as a flexible integrator in protecting other molecules under drought or other abiotic stress conditions [30,64].

We report, for the first time in chicory (*Cichorium intybus* L.), the partial amino acid sequence of *Ci*XTH29 (submission number: OQ572736), showing a high identity with *At*XTH29 (99.13%). RT-qPCR analyses revealed different basal *CiXTH29* transcription levels in the analyzed chicory plants; interestingly, the *CiXTH29* basal expression level was highest in the Selvatica and Zuccherina di Trieste varieties, followed, with low values, by Galatina. In response to a period of water shortage, no significant variations were observed in the *CiXTH29* expression level in the varieties with a high basal expression level of the gene, and only a slight, but statistically significant, increase was observed in Galatina variety. Only in Selvatica was a reduction of gene expression observed, but this was not statistically significant. On the contrary, *CiXTH29* expression was highly induced in the Brindisina, Esportazione and Rossa Italiana, varieties characterized by a low gene basal expression level. It would appear that the *CiXTH29* basal expression level is inversely correlated with its increased expression during water stress. The amount of *CiXTH29* basal expression in Galatina, even if higher than Brindisina, Esportazione and Rossa Italiana, could be not high enough to protect the plant from water stress damage. The high basal expression levels of *XTH29* in chicory varieties (Selvatica, Zuccherina di Trieste and Galatina) is certainly an important feature, since *XTH29* is known to be poorly expressed in *Arabidopsis* [65], in *Brassica* [66] and, as shown recently, in the roots and fruits of *Schima superba* [67]. Nevertheless, several pieces of evidence demonstrate an important role of XTHs in drought stress, as reported in *Medicago truncatula* [68], *Arabidopsis* [26,29], tomato [27], grapevine [69] and *Salicornia aeuropaea* [70]. Therefore, our data on *CiXTH29* can be added to the large body of evidence on the role of XTHs in drought stress response.

The constitutive expression level of LEA proteins is considered to be a marker of drought resistance, since more *LEA* genes are overexpressed in drought-resistant plants, as reported in the drought-resistant *Gossipium tomentosum* [71]. Significantly, the basal expression level of *CiLEA4*, evaluated in the analyzed chicory varieties, shows the same trend of *CiXTH29* expression, with a higher basal expression in the varieties that showed a higher resistance to drought stress. In rice, the drought-tolerant variety Aeron1 shows a high LEA basal expression, comparable to that of the *LEA*-overexpressing mutants MR219-4 and MR219-9. This suggests that *LEA* genes could serve as a potential tool for drought tolerance determination in rice [72]. In chicory, a particularly interesting feature is that the Selvatica, Zuccherina di Trieste and Galatina varieties show a basal expression of *CiLEA4* and *CiXTH29* that is higher than the other varieties. We, therefore, suggest that *CiLEA4* and *CiXTH29* gene expression could represent potential markers for drought stress tolerance in chicory.

In order to better understand the eventual reorganization of the xyloglucans in the apoplast of chicory varieties grown in hydrated or dehydrated soil, we immunolabeled transverse leaf midvein sections using the LM24 monoclonal antibody. The quantification of the fluorescent index gives indications of a general increase in the presence of xyloglucans in the cell wall of the bundle xylem vessels of almost all varieties. It is known that *AtXTH4* and *AtXTH9* contribute to wood cell expansion and secondary wall formation, altering the xyloglucan amount [73,74]. Notwithstanding, more in-depth investigations on the composition of the wall would be necessary to explain these results that seem to indicate, in almost all the analyzed chicory varieties, a reorganization of the cell wall, with an increase of the amount of xyloglucans as a response to a condition of water shortage, except in those varieties (Esportazione and Rossa Italiana) that are more susceptible to water stress.

## 5. Conclusions

In this study, we show evidence that some chicory varieties, having a high *CiXTH29* and *CiLEA4* basal expression, better tolerate moderate or severe drought stress conditions than varieties that overexpress the genes only in response to stress condition. The drought-tolerant chicory varieties (Selvatica and Zuccherina di Trieste) show a detectable basal expression of *CiXTH29* and *CiLEA4*, associated with a greater vegetative development of basal rosette leaves. When these two traits are not associated, the variety does not show the characteristic of resistance, as occurs for Brindisina (high basal rosette development, but low *CiXTH29* and *CiLEA4* expression) and Galatina (detectable *CiXTH29* and *CiLEA4* basal expression, but low vegetative development) varieties.

Our results are particularly significant for developing drought-resilience breeding strategies and selection programs of chicory varieties based on the XTH29 and LEA4 responses, using their corresponding genetic markers. Selected varieties may achieve desired plant stands even with limited humidity conditions for germination, and may tolerate prolonged drought stress periods during the crop development.

## Figures and Tables

**Figure 1 biology-12-00444-f001:**
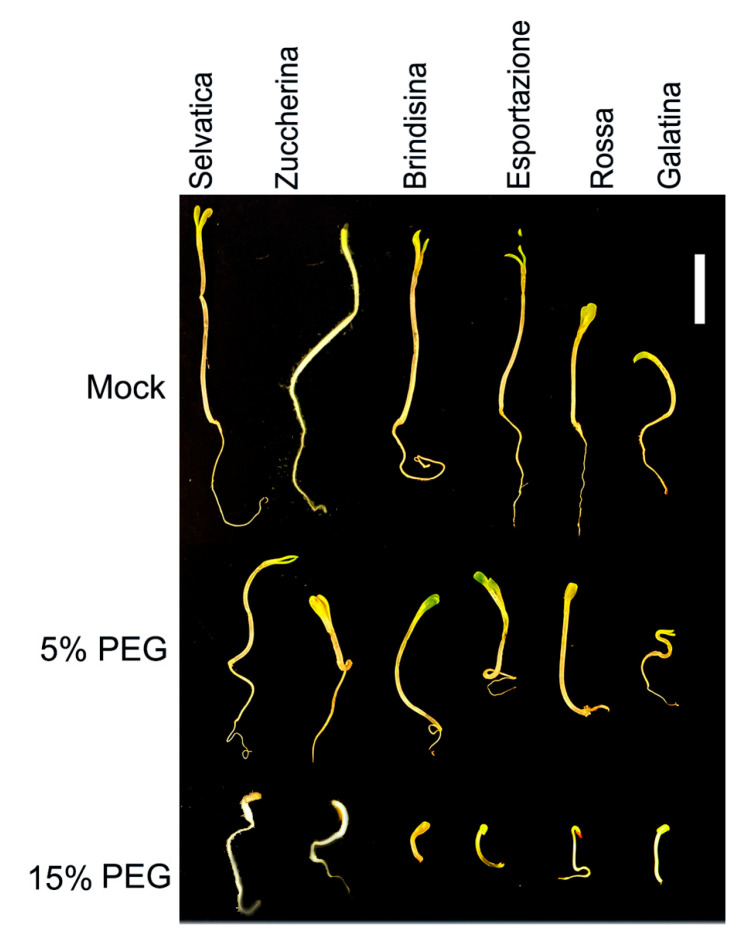
Six day-old chicory seedlings growing in distilled water (mock), as well as under moderate (5% PEG-6000 solution) and severe drought (15% PEG-6000 solution) conditions. Scale bar = 10 mm.

**Figure 2 biology-12-00444-f002:**
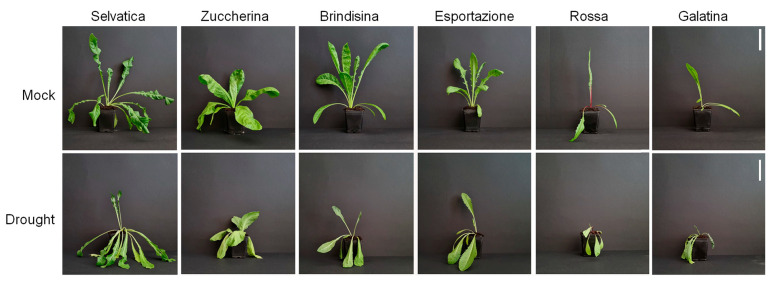
Phenotype of the analyzed chicory varieties. Six week-old chicory plants were grown, for a further 10 days, in hydrated soil (Mock) or without watering (Drought). Scale bar = 10 cm.

**Figure 3 biology-12-00444-f003:**
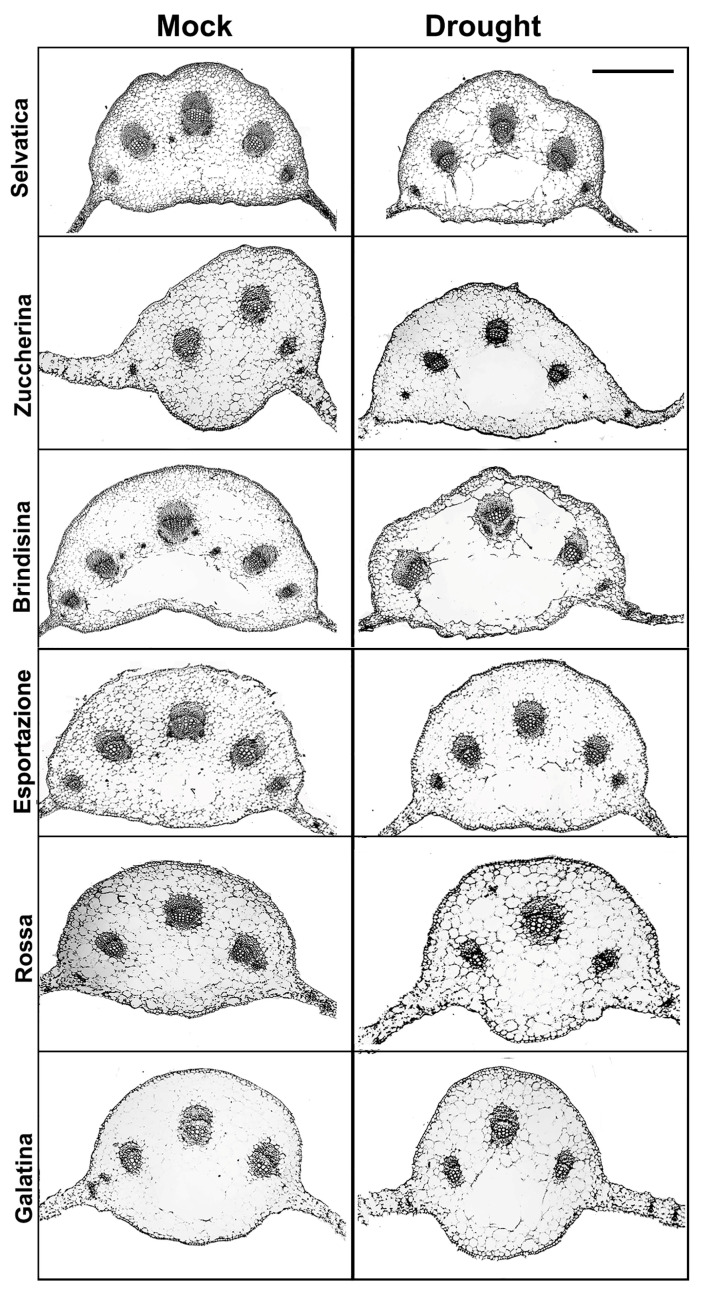
Brightfield images of transverse sections of the basal leaf central vein taken from 6 week-old chicory varieties grown, for a further 10 days, in hydrated soil (Mock) or without watering (Drought). Scale bar = 1 mm.

**Figure 4 biology-12-00444-f004:**
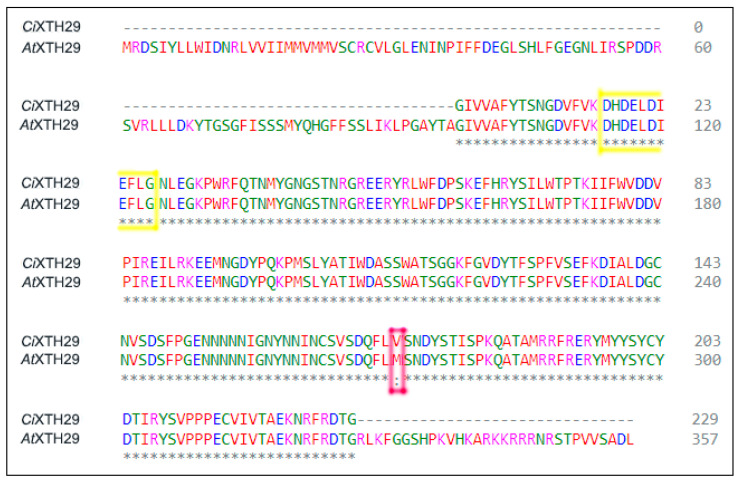
Comparison of *Ci*XTH29 partial amino acid sequence and the *At*XTH29 amino acid sequence. The yellow box indicates the catalytic site of the XTH family. The red box indicates a substitution in the *Ci*XTH29 amino acid sequence with respect to the *At*XTH29 one. “*” indicates identical amino acid residues, “:” indicates a conservative amino acid replacement.

**Figure 5 biology-12-00444-f005:**
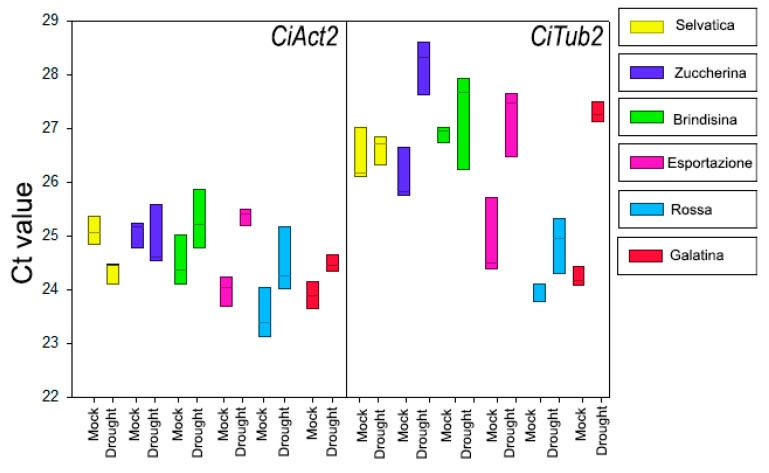
Ct values of the candidate references genes *CiAct2* and *CiTub2* in 6 week-old chicory plants kept for a further 10 days with (Mock) or without watering (Drought). The results are presented with box plots (middle bar, median; box limits, upper and lower quartiles; whiskers, min. and max. values).

**Figure 6 biology-12-00444-f006:**
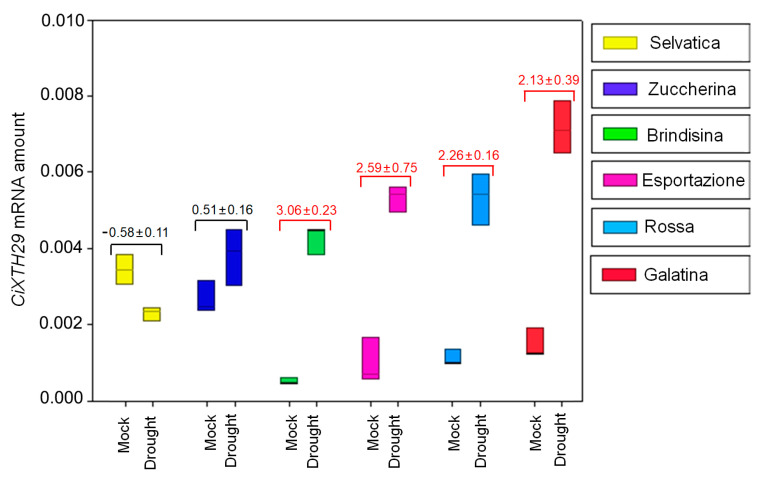
Expression pattern of the *CiXTH29* gene in basal rosette leaves of 6 week-old chicory plants kept for a further 10 days with (Mock) or without watering (Drought). Amplification output values of CiXTH29 mRNAs are expressed as 2^−∆Cq^, and are considered as being proportional to the amount of mRNA, according to [55]. The differential expression of *CiXTH29* in stressed conditions is reported as mean *±* standard deviation of log_2_FC, with respect to the control, for each variety. The red values indicate *CiXTH29’s* upregulation, the black values indicate no variation in the expression levels. The experiments were performed in three independent biological replicates. The results are presented with box plots (middle bar, median; box limits, upper and lower quartiles; whiskers, min. and max. values).

**Figure 7 biology-12-00444-f007:**
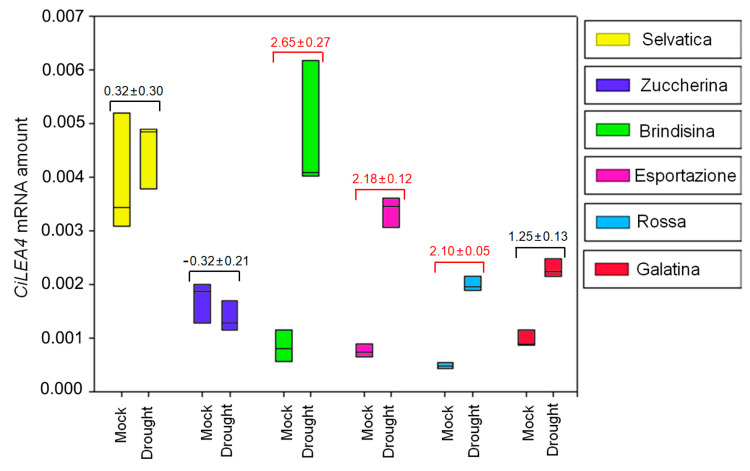
Expression pattern of *CiLEA4* gene in basal rosette leaves of 6 week-old chicory plants kept for a further 10 days with (Mock) or without watering (Drought). Amplification output values of *CiLEA4* mRNAs are expressed as 2^−∆Cq^, and are considered as being proportional to the amount of mRNA, according to [55]. The differential expression of *CiLEA4* in stressed conditions is reported as the mean *±* standard deviation of log_2_FC, with respect to the control, for each variety. The red values indicate *CiLEA4’s* upregulation, the black values indicate no variation in the expression levels. The experiments were performed in three independent biological replicates. The results are presented with box plots (middle bar, median; box limits, upper and lower quartiles; whiskers, min. and max. values).

**Figure 8 biology-12-00444-f008:**
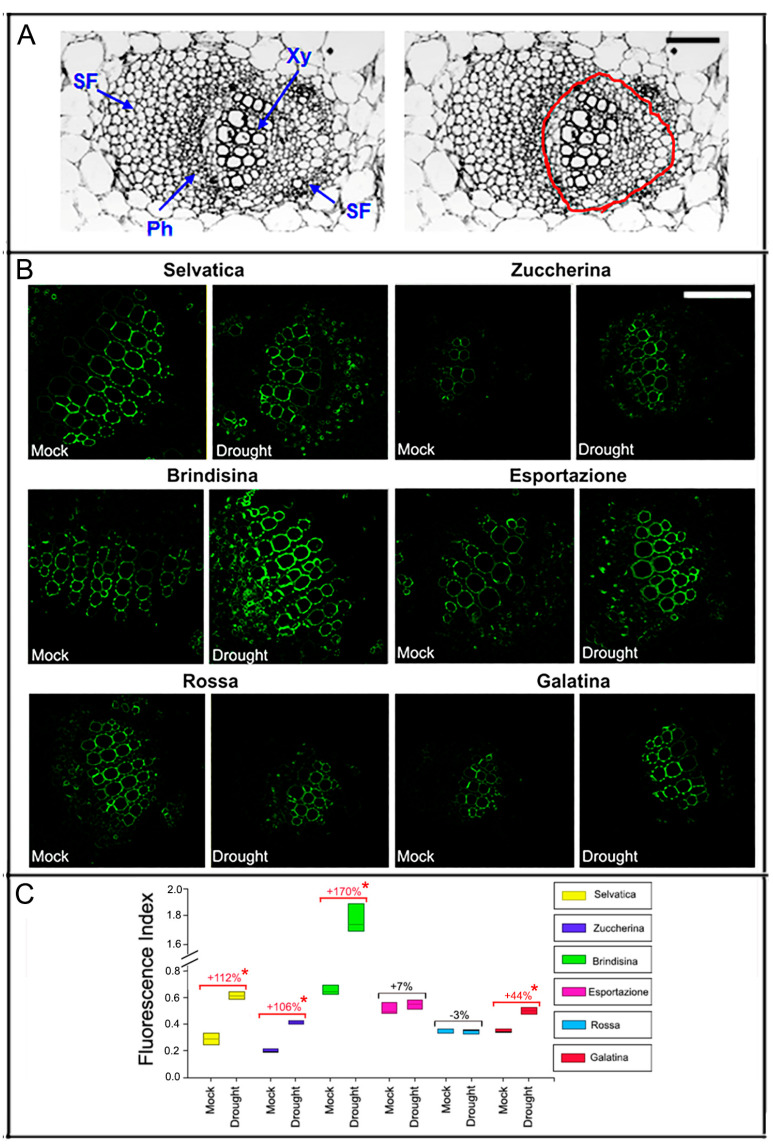
Detection of xyloglucans with LM24 xyloglucan monoclonal antibodies in the xylem area (red circle) of chicory leaves. (**A**) Brightfield image of the transverse section of the largest bundle of leaf midvein showing the area of the xylem used to calculate the xyloglucan content in 6 week-old chicory varieties grown, for a further 10 days, in hydrated soil (Mock) or without watering (Drought); the image refers to the Selvatica variety, reported as an example; (**B**) xyloglucans probed with LM24 antibody in the cell wall of xylem vessels. Scale bars = 100 μm (**A**,**B**). Scale bar = 100 µm. (**C**) Fluorescence index of the LM24 fluorescent antibody, quantified from confocal images; the percentage of fluorescence inhibition is also reported; the red values indicate statistically significant differences among Mock and Drought samples (* *p* < 0.001), according to the Student’s *t*-test. The results of three independent biological replicates are presented with box plots (middle bar, median; box limits, upper and lower quartiles; whiskers, min. and max. values). Xy, Xylem; Ph, Phloem; SF, Sclerenchyma Fiber.

**Table 1 biology-12-00444-t001:** Seed Germination Index (GI) of chicory varieties grown for 6 days in distilled water (Mock) or under drought stress, imposed by two different concentrations of PEG-6000. Data were submitted to a one-way analysis of variance (ANOVA); the asterisk indicates statistically significant differences among unstressed (Mock) and stressed samples (* *p* < 0.001), according to the Student’s *t*-test.

	Germination Index of Chicory Varieties (%)					
	Selvatica	Zuccherina	Brindisina	Esportazione	Rossa	Galatina
Mock	96.7 ± 2.2	90.0 ± 1.9	83.7 ± 3.3	80.0 ± 2.0	73.3 ± 2.4	60.1 ± 3.3
5% PEG-6000	96.7 ± 3.2	88.3 ± 2.2	61.7 ± 2.5 *	67.6 ± 3.2 *	68.1 ± 2.2 *	56.2 ± 2.1 *
15% PEG-6000	84.2 ± 3.2 *	76.7 ± 3.4 *	45.6 ± 2.4 *	59.8 ± 2.1 *	36.7 ± 1.7 *	49.7 ± 2.6 *

**Table 2 biology-12-00444-t002:** Comparative analyses of aerial part and root length values of 6 day-old chicory seedlings grown in distilled water (Mock) or under drought stress imposed by two different concentrations of PEG-6000. Data were submitted to a one-way analysis of variance (ANOVA): the asterisk indicates statistically significant differences among unstressed (Mock) and stressed samples (* *p* < 0.001), according to the Student’s *t*-test.

	Aerial Part Length (mm)					
	Selvatica	Zuccherina	Brindisina	Esportazione	Rossa	Galatina
Mock	28.95 ± 2.91	26.07 ± 5.32	20.59 ± 2.39	19.73 ± 1.74	16.80 ± 2.45	10.67 ± 1.63
5% PEG-6000	15.68 ± 3.90 *	11.00 ± 3.31 *	7.75 ± 1.06 *	7.10 ± 1.20 *	5.85 ± 2.03 *	5.00 ± 1.73 *
15% PEG-6000	5.71 ± 2.73 *	4.64 ± 1.28 *	2.93 ± 1.33 *	2.50 ± 1.17 *	1.71 ± 0.49 *	2.00 ± 0.71 *
	**Root length (mm)**					
	**Selvatica**	**Zuccherina**	**Brindisina**	**Esportazione**	**Rossa**	**Galatina**
Mock	22.79 ± 2.00	15.27 ± 2.22	20.55 ± 3.45	18.19 ± 2.76	11.53 ± 2.80	8.33 ± 1.37
5% PEG-6000	14.84 ± 2.63 *	7.94 ± 1.47 *	10.19 ± 2.07 *	8.06 ± 1.57 *	4.46 ± 2.73 *	4.54 ± 1.63 *
15% PEG-6000	5.71 ± 2.28 *	3.28 ± 1.14 *	2.20 ± 0.56 *	2.28 ± 2.13 *	1.34 ± 0.52 *	1.60 ± 0.55 *

## Data Availability

Data is contained within the article and Appendix A.

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
