# Peer review of "CiXTH29 and CiLEA4 Role in Water Stress Tolerance in Cichorium intybus Varieties"

_biology, 2023, doi:10.3390/biology12030444_

Round 1

Reviewer 1 Report

Dear Authors,

this is a well written manuscript, which unfortunately misses a clear statement of the hypothesis and objectives. At the same time, there are important results (gene expression of LEA4) that somehow were not introduced nor included in the conclusions. I do think that developing a well structured paragraph of hypothesis and objectives will help you (and the reader) to have a more focused manuscript. So please do so.

I reviewed the introduction, m&m, results and conclusion and provide you with some general and specific comments for your consideration. At this point, I would not review the discussion in depth as the manuscript needs some improvements.

Figures and Tables are informative (you might revise the titles and captions). I suggest replacing Figure 6 with a Table as I think can better display your results, but this is a decision for you to make.

I found a very well use of citations, which is very appreciated.

General comments:

-The simple summary (199 words) and the abstract (193 words) are similar and I do not see much value on having both as they are now. You either provide a more detailed abstract (~300 words, if appropriate with the journal) or a more succinct simple summary (<150 words) or you might modify both. As they stand now, I would suggest to drop the simple summary and edit the abstract.

-The general hypothesis is missing. You need to further develop paragraph in Ln89-93 providing the hypothesis and your general/specific objectives. It is not enough to tell what has been done but you need to explain what was the expected outcome of the research based on the state of art presented in your introduction. In my opinion, Ln91-93 are not appropriate because you are presenting your findings (‘...we identified specific morphological and genetic traits…’) which should be in results. Please, make sure to state which gene expression you studied (XTH29 alone or XTH29 and LEA4?), what are the treatments established (drought conditions in germination and in seedlings).

-Treatments: The 10 days un-irrigated treatment does not necessarily show a precise drought treatment. I wonder if you measured the water potential of those un-irrigated pots. If now, it could be useful to know at least how much water was provided to the irrigated pots or another way to measure the lack of water.

-Late embryogenesis abundant (LEA) proteins are presented in your results and well discussed, however, they are not mentioned in the introduction and unfortunately were totally neglected in the conclusions. These results seems as important as the ones related to XTH29. I think you need to decide whether to include them in your manuscript (and accordingly modify the introduction, abstract and conclusions) or to drop them completely to include in another manuscript.

-I have the feeling that the introduction needs some more ‘food’ to actually lead to a well presented hypothesis and objective. However, this is just a personal impression.

-In your results, you could start explaining what you have done (e.g., ‘We investigated the gene expression of ….because...REFERENCES’. However, this is usually done in your introduction and if you need to add more details here is because they are missing in the introduction section. The results sections is meant to present results rather than explaining why you did what. These explanations should be placed in your introduction and might also be included in your discussion if you are contrasting your results with others. Thus, sentence in Ln183-184 should be deleted.

-Section 3.1 present results of XTH29 and LEA4 genes, however, LEA4 has not been introduced before and the results are missing in the abstract, etc.

-Please, as much as possible, always follow the same order along the manuscript. For example, in the methods you described first the germination treatment, then the treatment in pots. Therefore, you should also first present the results in germination (section 3.2) and then the results in the potted plants (section 3.1).

-I would really consider to re-structure how results are presented. For example, the morphological assessment is a great visual assessment of the ‘drought’ treatment and could be presented at the very beginning to show the effects (only visually) of your treatment. Then, this can be followed by the gene expression results, which were actually measured and represent the most important results. Again, I leave that to you to decide according to your own writing style.

Specific comments:

-Ln17. Please, delete ‘the’ in: ...we investigate the response to drought…

-Ln17-19. Please, use brackets and, italicize and add a dot to ‘i.e.,’ when referring to the varieties: ‘Cichorium intybus L. (i.e., Selvatica, Zuccherina…). You may also replace ‘of different varieties’ with ‘of six varieties’. You may also consider to rephrase: ‘...to drought stress of six varieties...(i.e., ) with focus on the XTH29 gene, which is known as controlling plant stress response(s).’

Ln20-22. You may rephrase: ‘...we show evidence that chicory varieties with high CiXTH29 basal expression and vegetative development better tolerate drought stress conditions than varieties with over expression of the gene only in response to drought.’

-Ln90. Please, italicize and add the corresponding dot in ‘i.e.,’. Revise the whole manuscript.

-Ln103-104. I suggest to split the sentence here: ‘...two layers of Whatman No. 1 filter paper (Whatman Ltd., UK). Petri dishes were embedded with 5 ml of distilled water (control), 5% solution of PEG-6000 or 15% PEG-6000 solution to create germination drought conditions [REFERENCE]’. Please, add a reference here justifying these drought level conditions. Reference 34 allows to calculate the osmotic potential of the PEG solutions but you need to justify why you choose the selected levels of ‘drought’, which you later in the discussion refer to moderate and severe.

-Ln144. Suggest ‘replicates per variety and treatment’.

-Ln154. Please, double check if there is a more appropriate reference than Chen et al. [37] to consider gene expression changes as significant. You should probably find a more methodological publication.

-Ln161. It seems there is a double space: ‘described in De Caroli et al.’.

-L182. After reading the introduction, it is a surprise to read about the expression of CiLEA4. This has not been mentioned in the introduction nor justified its study. Please, consider whether this is appropriate in your manuscript and if so, then add the corresponding paragraph into your introduction.

-L197-198. It seems that the protein sequence has been described here for first time (that’s why it will be deposited in NCBI) and you need to highlight this. Isn’t it a result in itself?

-Ln244. Please, replace ‘e’ with ‘and’ in: ‘in Selvatica e Zuccherina di Trieste varieties’.

-Ln242-248. These results refer to the gene expression in mock conditions for all varieties, therefore, I suggest to rephrase: ‘In mock conditions, the RT-qPCR analysis revealed highest CiXTH29 expression levels in Selvatica and Zuccherina di Trieste (…), intermediate values in Rossa Italiana and Galatina (…) and the lowest in Brindisima and Exportazione (…); although, the latter had a large variation that also overlapped with gene expression observed in Rossa and Galatina varieties.’

-Ln248-254. I suggest to rephrase to something like this: ‘The 10-days drought treatment showed slight changes in the CiXTH29 expression in Selvatica and Zuccherina di Trieste (…), while significant changes (WHAT IS SIGNIFICANT?) were observed in Brindisima (…), Exportazione (…) and Rossa Italiana (…). Note that Selvatica was the only variety that reduced the gene expression after drought.’.

-Ln255-259. This is not appropriate in the results section. Please, delete and only present results in this section. I would start this paragraph with: ‘Similarly to CiXTH29, the expression pattern for CiLEA4 showed…’.

-Ln292-295. Please, summarize this to ‘In mock conditions, the average GI for all varieties was XX% (ranging from 97% to 60%), which dropped to XX% with 5% PEG solution and to XX% with 15% PEG solution (Table 1). Selvatica and Zuccherina varieties showed significant decrease of GI with the 15% PEG solution, while all other varieties significantly decreased their GI with both, the 5 and 15% solutions.

-Ln295-307: this can and should be summarize, you don’t need to read the entire table in the text, but highly the results shown in the table.

-Ln304-307. This statement is either confusing or incorrect.

-Ln313-324. This paragraph summarize well your results and this is how you should summarize the results of Table 1. You may consider to present the actual results described in this paragraph in a Table showing root and aerial growth (in mm) for the three treatments in rows and the six varieties in columns (similar to Table 1) instead of Figure 6, which unfortunately is less friendly to read.

-Figure 5 is an excellent visual representation of the results. Please, consider to rephrase the title to: ‘Six-day-old chicory seedlings growing in distilled water (mock), under moderate drought (5% PEG-6000 solution) and high drought (15% PEG-6000 solution). Scale bar = 10 mm.’

-Ln382. I think the following statement is incorrect and might be misleading: ‘Significantly, in all chicory varieties, the xyloglucans…’. As per the results in Fig. 9C, varieties Esportazione and Rossa changed 7% and 3% and may not represent a significant (statistically significant) difference. Please, revise your results and correct this whole results section (3.4).

-Ln578-580. This sentence seems inappropriate for conclusions and I suggest to delete it. You might start this second paragraph as: ‘Our results could be particularly useful to develop drought-resilience breeding strategies and for the genetic selection of chicory varieties…’.

I hope these comments are useful!!!

Author Response

Dear Authors,

this is a well written manuscript, which unfortunately misses a clear statement of the hypothesis and objectives. At the same time, there are important results (gene expression of LEA4) that somehow were not introduced nor included in the conclusions. I do think that developing a well structured paragraph of hypothesis and objectives will help you (and the reader) to have a more focused manuscript. So please do so.

I reviewed the introduction, m&m, results and conclusion and provide you with some general and specific comments for your consideration. At this point, I would not review the discussion in depth as the manuscript needs some improvements.

Figures and Tables are informative (you might revise the titles and captions). I suggest replacing Figure 6 with a Table as I think can better display your results, but this is a decision for you to make.

I found a very well use of citations, which is very appreciated.

Basically, we have followed all precious technical suggestions and English style corrections given by referee number 1. Of course, we would like to thank very much the unknown referee for having taken great care of our manuscript. However, we have reported our corrections in the manuscript that are marked up using the “Track Changes” function. Now the manuscript has been modified in all sections of it. Thanks very much again. 

Overall, the statement and objective of the study have been clarified, and more information about LEA proteins have been provided in the Introduction, Materials and Methods, and Discussion.  We have revised the captions of the figures as suggested and replaced the Figure 6 in Table 2.

General comments:

-The simple summary (199 words) and the abstract (193 words) are similar and I do not see much value on having both as they are now. You either provide a more detailed abstract (~300 words, if appropriate with the journal) or a more succinct simple summary (<150 words) or you might modify both. As they stand now, I would suggest to drop the simple summary and edit the abstract.

The instructions for authors of the Biology journal require the simple summary and the abstract, we do have shorted the simple summary and modified the abstract as you suggested.

-The general hypothesis is missing. You need to further develop paragraph in Ln89-93 providing the hypothesis and your general/specific objectives. It is not enough to tell what has been done but you need to explain what was the expected outcome of the research based on the state of art presented in your introduction. In my opinion, Ln91-93 are not appropriate because you are presenting your findings (‘...we identified specific morphological and genetic traits…’) which should be in results. Please, make sure to state which gene expression you studied (XTH29 alone or XTH29 and LEA4?), what are the treatments established (drought conditions in germination and in seedlings).

We have change sentence 91-93 and modify the Introduction as suggested.

-Treatments: The 10 days un-irrigated treatment does not necessarily show a precise drought treatment. I wonder if you measured the water potential of those un-irrigated pots. If now, it could be useful to know at least how much water was provided to the irrigated pots or another way to measure the lack of water.

We were not able to measure the water potential, but we have provided the amount of water used for irrigation and how often it was administered.

-Late embryogenesis abundant (LEA) proteins are presented in your results and well discussed, however, they are not mentioned in the introduction and unfortunately were totally neglected in the conclusions. These results seems as important as the ones related to XTH29. I think you need to decide whether to include them in your manuscript (and accordingly modify the introduction, abstract and conclusions) or to drop them completely to include in another manuscript.

We have inserted more details about LEA proteins in all manuscript sections.

-I have the feeling that the introduction needs some more ‘food’ to actually lead to a well presented hypothesis and objective. However, this is just a personal impression.

We have added more “food” information about chicory.

-In your results, you could start explaining what you have done (e.g., ‘We investigated the gene expression of ….because...REFERENCES’. However, this is usually done in your introduction and if you need to add more details here is because they are missing in the introduction section. The results sections is meant to present results rather than explaining why you did what. These explanations should be placed in your introduction and might also be included in your discussion if you are contrasting your results with others. Thus, sentence in Ln183-184 should be deleted.

-Section 3.1 present results of XTH29 and LEA4 genes, however, LEA4 has not been introduced before and the results are missing in the abstract, etc.

-Please, as much as possible, always follow the same order along the manuscript. For example, in the methods you described first the germination treatment, then the treatment in pots. Therefore, you should also first present the results in germination (section 3.2) and then the results in the potted plants (section 3.1).

-I would really consider to re-structure how results are presented. For example, the morphological assessment is a great visual assessment of the ‘drought’ treatment and could be presented at the very beginning to show the effects (only visually) of your treatment. Then, this can be followed by the gene expression results, which were actually measured and represent the most important results. Again, I leave that to you to decide according to your own writing style.

We have followed all the suggested indications.

Specific comments:

-Ln17. Please, delete ‘the’ in: ...we investigate the response to drought…

-Ln17-19. Please, use brackets and, italicize and add a dot to ‘i.e.,’ when referring to the varieties: ‘Cichorium intybus L. (i.e., Selvatica, Zuccherina…). You may also replace ‘of different varieties’ with ‘of six varieties’. You may also consider to rephrase: ‘...to drought stress of six varieties...(i.e., ) with focus on the XTH29 gene, which is known as controlling plant stress response(s).’

Ln20-22. You may rephrase: ‘...we show evidence that chicory varieties with high CiXTH29 basal expression and vegetative development better tolerate drought stress conditions than varieties with over expression of the gene only in response to drought.’

-Ln90. Please, italicize and add the corresponding dot in ‘i.e.,’. Revise the whole manuscript.

-Ln103-104. I suggest to split the sentence here: ‘...two layers of Whatman No. 1 filter paper (Whatman Ltd., UK). Petri dishes were embedded with 5 ml of distilled water (control), 5% solution of PEG-6000 or 15% PEG-6000 solution to create germination drought conditions [REFERENCE]’. Please, add a reference here justifying these drought level conditions. Reference 34 allows to calculate the osmotic potential of the PEG solutions but you need to justify why you choose the selected levels of ‘drought’, which you later in the discussion refer to moderate and severe.

-Ln144. Suggest ‘replicates per variety and treatment’.

-Ln154. Please, double check if there is a more appropriate reference than Chen et al. [37] to consider gene expression changes as significant. You should probably find a more methodological publication.

-Ln161. It seems there is a double space: ‘described in De Caroli et al.’.

-L182. After reading the introduction, it is a surprise to read about the expression of CiLEA4. This has not been mentioned in the introduction nor justified its study. Please, consider whether this is appropriate in your manuscript and if so, then add the corresponding paragraph into your introduction.

-L197-198. It seems that the protein sequence has been described here for first time (that’s why it will be deposited in NCBI) and you need to highlight this. Isn’t it a result in itself?

-Ln244. Please, replace ‘e’ with ‘and’ in: ‘in Selvatica e Zuccherina di Trieste varieties’.

-Ln242-248. These results refer to the gene expression in mock conditions for all varieties, therefore, I suggest to rephrase: ‘In mock conditions, the RT-qPCR analysis revealed highest CiXTH29 expression levels in Selvatica and Zuccherina di Trieste (…), intermediate values in Rossa Italiana and Galatina (…) and the lowest in Brindisima and Exportazione (…); although, the latter had a large variation that also overlapped with gene expression observed in Rossa and Galatina varieties.’

-Ln248-254. I suggest to rephrase to something like this: ‘The 10-days drought treatment showed slight changes in the CiXTH29 expression in Selvatica and Zuccherina di Trieste (…), while significant changes (WHAT IS SIGNIFICANT?) were observed in Brindisima (…), Exportazione (…) and Rossa Italiana (…). Note that Selvatica was the only variety that reduced the gene expression after drought.’.

-Ln255-259. This is not appropriate in the results section. Please, delete and only present results in this section. I would start this paragraph with: ‘Similarly to CiXTH29, the expression pattern for CiLEA4 showed…’.

-Ln292-295. Please, summarize this to ‘In mock conditions, the average GI for all varieties was XX% (ranging from 97% to 60%), which dropped to XX% with 5% PEG solution and to XX% with 15% PEG solution (Table 1). Selvatica and Zuccherina varieties showed significant decrease of GI with the 15% PEG solution, while all other varieties significantly decreased their GI with both, the 5 and 15% solutions.

-Ln295-307: this can and should be summarize, you don’t need to read the entire table in the text, but highly the results shown in the table.

-Ln304-307. This statement is either confusing or incorrect.

-Ln313-324. This paragraph summarize well your results and this is how you should summarize the results of Table 1. You may consider to present the actual results described in this paragraph in a Table showing root and aerial growth (in mm) for the three treatments in rows and the six varieties in columns (similar to Table 1) instead of Figure 6, which unfortunately is less friendly to read.

-Figure 5 is an excellent visual representation of the results. Please, consider to rephrase the title to: ‘Six-day-old chicory seedlings growing in distilled water (mock), under moderate drought (5% PEG-6000 solution) and high drought (15% PEG-6000 solution). Scale bar = 10 mm.’

-Ln382. I think the following statement is incorrect and might be misleading: ‘Significantly, in all chicory varieties, the xyloglucans…’. As per the results in Fig. 9C, varieties Esportazione and Rossa changed 7% and 3% and may not represent a significant (statistically significant) difference. Please, revise your results and correct this whole results section (3.4).

-Ln578-580. This sentence seems inappropriate for conclusions and I suggest to delete it. You might start this second paragraph as: ‘Our results could be particularly useful to develop drought-resilience breeding strategies and for the genetic selection of chicory varieties…’.

I hope these comments are useful!!!

Your comments have been very helpful and constructive, we have rephrased and summarized all the indicated sentences and the results have been rearranged as suggested.

Reviewer 2 Report

biology-2223506-peer-review-v1

 Article:  CiXTH29 expression and water stress tolerance in Cichorium intybus varieties

The idea of the manuscript is very modest, in general, the measurements are insufficient to achieve the objective.

Abstracts: good.

Keywords: remove "drought tolerance traits"

Introduction:

In general, the introduction lacks a paragraph related to the negative effect of drought stress on the growth stages of chicory plants as well as gene expression.

Materials and Methods:

Line 96: remove " and plant growth "

The authors used polyethylene glycol 6000 as a drought stress treatment and inferred this from reference (34), what is the relationship of this PEG to drought stress?

The authors neglected the design of the experiments in this study.

Line 106-107: The experiment was repeated 5 times, 106 with a total of 50 seeds for variety, Please clarify what is meant?

Results:

Line 183: the author's open the results section with a reference discussing the effect of heat and drought stress in Arabidopsis seedlings, I suggest moving this paragraph to the discussion part.

Line 204-209: this paragraph presents information related to the materials and methods part, so it must be explained in detail in this part.

Line 255: LEA (Late Embryogenesis Abundant), the authors did not refer to it in the materials and methods section.

Line 255:266:  in this paragraph, many references have been used. It is preferable to focus on that in the discussion part.

3.3. Phenotypic analysis of chicory plants under 10 days of dehydration, unfortunately, this paragraph does not add anything new to the manuscript.

Discussion:

The discussion is totally inconsistent with the simple measurements of the results part, and is too long.

Conclusions:

Line 578: 583: paragraph preferably moved to the introduction part.

Author Response

biology-2223506-peer-review-v1

 Article:  CiXTH29 expression and water stress tolerance in Cichorium intybus varieties

The idea of the manuscript is very modest, in general, the measurements are insufficient to achieve the objective.

Abstracts: good.

Keywords: remove "drought tolerance traits"

Introduction:

In general, the introduction lacks a paragraph related to the negative effect of drought stress on the growth stages of chicory plants as well as gene expression.

Materials and Methods:

Line 96: remove " and plant growth "

The authors used polyethylene glycol 6000 as a drought stress treatment and inferred this from reference (34), what is the relationship of this PEG to drought stress?

The authors neglected the design of the experiments in this study.

Line 106-107: The experiment was repeated 5 times, 106 with a total of 50 seeds for variety, Please clarify what is meant?

Results:

Line 183: the author's open the results section with a reference discussing the effect of heat and drought stress in Arabidopsis seedlings, I suggest moving this paragraph to the discussion part.

Line 204-209: this paragraph presents information related to the materials and methods part, so it must be explained in detail in this part.

Line 255: LEA (Late Embryogenesis Abundant), the authors did not refer to it in the materials and methods section.

Line 255:266:  in this paragraph, many references have been used. It is preferable to focus on that in the discussion part.

3.3. Phenotypic analysis of chicory plants under 10 days of dehydration, unfortunately, this paragraph does not add anything new to the manuscript.

Discussion:

The discussion is totally inconsistent with the simple measurements of the results part, and is too long.

Conclusions:

Line 578: 583: paragraph preferably moved to the introduction part.

According to all your precious indications above reported, we have done the suggested modifications visible in the manuscript using the “Track Changes” function. Referring to the point lines: 204-209, we have kept the paragraph in the Results since it is the first identification of a reference gene in the six chicory varieties under drought stress conditions. The discussion has been shortened as suggested.

Reviewer 3 Report

1- Short simple summary.

2- Add and explain the number of traits as quantitative that were measured in the experiment in the abstract.

3- Add the references for each part of the materials and methods.

4- Why did you choose these varieties?

5- Where are the references for the drought method?

6- Why did you choose the method of Chen's et al, 2007, for measuring the expression genes? it is not a famous formula for gene expression. 

The best and most common formula is livak and schmittgen (2001).

Author Response

Thanks very much for having taken care of our manuscript and for your suggestions on modifying it. We report below our answer.

We have shorted the simple summary and modified the abstract as suggested.

We have added the lacking reference in Materials and Methods.

We have chosen a wild (Selvatica) variety and some of the most cultivated chicory varieties for food use.

We have added the reference that you have asked.

Method of Chen's et al, 2007 was mentioned for Log2FoldChange formula, then we have changed Livak and Schmittgen (2008) with Livak and Schmittgen (2001), as you suggested, when relative gene expression was reported.

Reviewer 4 Report

Review of the manuscript written by Monica De Caroli and colleagues entitled "CiXTH29 expression and water stress tolerance in Cichorium intybus varieties". The manuscript shows how chicory varieties differ in their tolerance to dehydration. These are interesting studies, in terms of both subject and logically planned experiments. The authors demonstrated an important role of XTH29 in dehydration tolerance.

I just have a note - a question for the authors regarding the selection of the age of the plants. The six-week-old chicory varieties, Rosa and Galatina, were smaller than the other varieties. They appear to be at a lower stage of ontogenesis and are younger in development. They have only three vascular bundles in their leaves, while the other varieties have more vascular bundles. It is possible that this difference in plant development affected the results of this study. Have the authors tried to see if 7-8 week old plants have more of these bundles? Perhaps, it was necessary to choose plants of other ages for research, with a similar level of advancement in their development.

Author Response

Thank you very much for your observations about the age of chicory varieties used in our study. However, with 7- or 8-week-old plants the pattern growth is more or less the same. So, we decided to perform the experiment using 6-week-old plant considering that our aim was mainly related to study the tolerance to drought stress of the six chosen chicory varieties.

Round 2

Reviewer 1 Report

Dear Authors,

In this manuscript, the authors study the genetic expression of CiXTH29 and CiLEA4 in six chicory varieties in response to drought treatments in two stages, germination (6 days) and during crop development (6-weeks plants). The results are very comprehensive and provide great morphological detail of the drought response that clearly reflect the effects of the treatments. Then, the genetic response under investigation is also well presented showing the relevance of the selected genes in the chicory response to drought.

I am delighted seen the improvement of the manuscript. Please, consider to incorporate the following comments that I hope will improve it.

GENERAL COMMENTS

I think it is a good practice to clearly state your hypothesis in the introduction.

Since the title refers to the role of CiXTH29 and CiLEA4, it seems important to give a conclusion on this aspect.

The use of references is almost excellent (I am very thankful to that), consider the couple of comments given. Very good use of references, excellent written style and excellent general structure of the manuscript.

SPECIFIC COMMENTS

Keywords: Please, delete those keywords that are already used in the title (e.g., Cichorium intybus, etc.) and replace them with other. In general, the most important words should be included in the title, and second most important words can be added as keywords.

Please, add a coma, after using i.e., and e.g.,

Ln41. Please, consider replacing ‘agriculture field’ with ‘rainfed agriculture’

Ln203. Please, delete ‘and’ in: “…; furthermore, AND the reduced”

Ln205. Please, delete the references [15] and [16] that are not relevant for chicory and replace for appropriate references.

Ln206. Please, replace ‘show’ with ‘showed’: ‘..to rainfed condition stress showed an increase…’

Ln211-213. This sentence is correct, yet, I wonder if it would be better to clearly express here what was the expected response to the induced drought. I recommend to add here your general hypothesis and main objectives, then, you could or not describe what has been done.

-Please, confirm that a level of significance of <0.001 instead of the most commonly used <0.01 was used in your analysis. Otherwise, please, revise the document accordingly. Ln386, Ln411, etc.

Ln1210-1212. You may provide a reference for this statement

Ln1218-1220. Please, replace ‘we have previously reported’ with ‘we have previously mentioned’ or something like that.

Discussion. You should consider to add subsections within your discussion. For example, Germination and morphological response for Ln1139-1213, Cell wall changes of XTHs for Ln1214-1246, Cell wall changes of LEA proteins for Ln1247-1259.

Ln1260-1972 (please, double check the line numbering in page 17). These are interesting results and it is acceptable to mention the need of further research to understand them. I wonder if there is any publication that mention such phenomena. If not, you could state that to your knowledge, this cell reorganizations has not been reported previously.

Conclusion

Following your title: what is the role of XTH29 and LEA in drought tolerance of chicory? You may need to say something in this section.

Ln1983-1986. Please, consider to correct to: ‘...significant for developing drought-resilience breeding strategies and selection programs of chicory varieties based on the XTH29 and LEA responses using their corresponding genetic makers. Selected varieties may achieve desired plant stands even with limited humidity for germination and may tolerate prolonged drought stress periods during the crop development’.

Author Response

All the authors are grateful to the reviewer for the precious work that has significantly improved our manuscript. We have followed all reviewer’s suggestions, with the exception of adding subsections in the discussion. We prefer to keep the discussion as is it, because the two aspects (morphology and gene expression) are correlated in our thoughts. 

Reviewer 2 Report

Most of concerns have been addressed in this version. In my opinion, this manuscript can be accepted in this form

Author Response

All the authors are grateful to the reviewer for the work that has significantly improved our manuscript. 

Reviewer 3 Report

Accept 

Author Response

(The authors gave the same response as above.)
